# Extremely Robust Remote-Target Detection Based on Carbon Dioxide-Double Spikes in Midwave Spectral Imaging

**DOI:** 10.3390/s20102896

**Published:** 2020-05-20

**Authors:** Sungho Kim, Jungsub Shin, Joonmo Ahn, Sunho Kim

**Affiliations:** 1Department of Electronic Engineering, Yeungnam University, 280 Daehak-Ro, Gyeongsan, Gyeongbuk 38541, Korea; 2Agency for Defense Development, P.O. Box 35, Daejeon 34186, Korea; jss@add.re.kr (J.S.); ahnjm@add.re.kr (J.A.); edl423@add.re.kr (S.K.)

**Keywords:** ship detection, false alarm, farbon dioxide peaks, midwave infrared, FTIR

## Abstract

Infrared ship-target detection for sea surveillance from the coast is very challenging because of strong background clutter, such as cloud and sea glint. Conventional approaches utilize either spatial or temporal information to reduce false positives. This paper proposes a completely different approach, called carbon dioxide-double spike (CO_2_-DS) detection in midwave spectral imaging. The proposed CO_2_-DS is based on the spectral feature where a hot CO_2_ emission band is broader than that which is absorbed by normal atmospheric CO_2_, which generates CO_2_-double spikes. A directional-mean subtraction filter (D-MSF) detects each CO_2_ spike, and final targets are detected by joint analysis of both types of detection. The most important property of CO_2_-DS detection is that it generates an extremely low number of false positive caused by background clutter. Only the hot CO_2_ spike of a ship plume can penetrate atmosphere, and furthermore, there are only ship CO_2_ plume signatures in the double spikes of different spectral bands. Experimental results using midwave Fourier transform infrared (FTIR) in a remote sea environment validate the extreme robustness of the proposed ship-target detection.

## 1. Introduction

Remote-ship detection is important in various applications, such as maritime navigation [1], coast guard searches [2], and homeland security procedures [3]. Infrared images are frequently adopted, especially due to their operational capability, day or night.

The key issue is how to reduce the high number of false detections caused by cloud clutter and sea surface glint while maintaining an acceptable detection rate [4,5]. Cloud edges produce false detections, and sea glint is similar to small infrared targets, both of which degrade detection performance. Since the 1990s, a lot of methods have been proposed to reduce the false positives by using either spatial information or temporal information from infrared images.

Spatial information processing, such as background subtraction, can be a feasible approach to reducing background clutter. Background images are estimated by spatial filters, such as the mean filter [6], the least mean square filter [7,8], the median filter [9], and the morphological filter [10]. These filters use neighboring pixels to estimate background pixel values, but with different strategies. In particular, mean subtraction filter (MSF)-based target detection is the most simple, but is weak when it comes to cloud edges. Non-linear filters, such as max-median, morphology, and data-fitting, show better cloud clutter suppression capabilities [11,12]. Local directional Laplacian-of-Gaussian filtering can remove false positives from cloud edges [13]. In the spatial information approach, classification with a spatial shape feature can discriminate clutter. Hysteresis thresholding [14], statistics-based adaptive thresholding [15], the Bayesian classifier [16], and support vector machines [17] are well-known methods. Voting by various classifiers can enhance dim-target detection rates [18]. Spatial frequency information can be used to remove low-frequency clutter, such as the three-dimensional fast Fourier transform (3D-FFT) spectrum [19]. The wavelet transform is effective against sea glint [20], and low pass filter can also deal with sea glint [21]. An adaptive high-pass filter can reduce cloud and sea glint clutter [22], while spatial target-background contextual information enhances the target signature and reduces background clutter, which is effective for sea glint reduction [23]. Spatial multi-feature fusion can increase the clutter discrimination capability [24].

Temporal information processing, such as track-before-detection, can remove slowly moving cloud clutter and fast-blinking sea glint [25]. Temporal profiles are useful to discriminate between a fast-moving target and slow-moving cloud [26,27,28]. The temporal contrast filter is suitable for detecting small, supersonic, infrared targets [29], and a 3D matched filter for wide-to-exact searches can remove cloud clutter very fast [30]. A power-law detector for image frames is effective for target detection in dense clutter [31]. Since a previous frame can be regarded as background, a weighted autocorrelation matrix update using the recursive technique can be useful in order to eliminate sea glint [32]. An advanced adaptive spatial-temporal filter can achieve tremendous gain [33], and principal component analysis in multi-frames can alleviate false positives from sun glint [34].

Synthetic aperture radar (SAR) with deep learning can be a useful approach for remote ship detection application [35,36,37]. Faster R-CNN [35], DRBox [36], and Contextual CNN [37] can provide improved detection performance but these approaches require huge number of training dataset to reduce false positives.

As mentioned above, the conventional approaches in the spatial- or temporal-image domains have their own pros and cons, depending on the situations and conditions. The spatial filter based approach relies on target shapes and intensity distributions, which is ambiguous if targets are far away. The temporal filter based approach assumes fast target motion with stationary sensors. If this assumption is not satisfied, it will generate many false positives in the maritime environment.

This paper proposes a novel spectral–spatial signature analysis-based ship detection method in midwave hyperspectral images, instead of the conventional spatial or temporal methods. Based on the combustion process, CO_2_ emissions show a double-spike signature at around 4.16 μm (2400 cm^−1^) and 4.34 μm (2300 cm^−1^). The hot CO_2_ emission band is broader than that which is absorbed by normal atmospheric CO_2_, which generates CO_2_-double spikes. The directional-mean subtraction filter (D-MSF) detects each CO_2_ spike, and final targets are detected by joint analysis of the detection of both spikes. So, we call the proposed method carbon dioxide-double spike (CO_2_-DS) detection.

The contributions from this paper can be summarized as follows. First, the phenomenon of the CO_2_-double spike emission in the midwave band is analyzed systematically. Second, a novel spectral–spatial analysis is proposed to detect remote ships in the maritime environment based on this analysis. Third, CO_2_-DS detection can suppress background clutter (cloud, sea glints) completely, which leads to an extremely low false positive rate in remote-ship detection. Finally, the performance of CO_2_-DS detection is demonstrated in a real sea environment with real ships.

The remainder of this paper is organized as follows. Section 2 explains the proposed CO_2_-DS method, including analysis of carbon dioxide-double spike radiation. Section 3 evaluates the clutter suppression performance of the CO_2_-DS method in the maritime environment. The paper concludes in Section 4.

## 2. Proposed Ship CO_2_ Plume Detection

### 2.1. Signature Analysis of Carbon Dioxide-Double Spikes

Before explaining the details of the proposed CO_2_-DS, the double-spike phenomenon in midwave spectral bands should be introduced [38,39]. Figure 1 summarizes the overall mechanism of CO_2_-double spikes in received spectral radiance from ship plumes. Hot CO_2_ emits thermal energy in wider spectral band than that of normal air CO_2_, as shown in Figure 1a. Atmospheric spectral transmittance shows strong absorption at 2300–2380 cm^−1^ by normal CO_2_ in the atmosphere, as shown in Figure 1b. Therefore, the received spectral radiance in a TELOPS Fourier transform infrared (FTIR) camera [40] shows a double-spike signature: the first spike is around 2276 cm^−1^ (4.39 μm), and the second spike is around 2393 cm^−1^ (4.18 μm).

The measurement of CO_2_-double spikes can be derived from radiative transfer calculated with Equation (Equation 1). We adopted the radiative-transfer equation used in the Moderate-Resolution Atmospheric Radiance and Transmittance (MODTRAN) [41]. In general, at-sensor received radiance in the midwave infrared (MWIR) region consists of CO_2_-emitted radiance, transmitted background radiance, and total atmospheric path radiance (thermal+solar components).
(1)$$\begin{array}{c} L_{target}\left(\lambda \right)=\tau \left(\lambda \right)\left[\varepsilon \left(\lambda \right)B_{CO_{2}}\left(\lambda ,T_{CO_{2}}\right)+\left(1-\varepsilon \left(\lambda \right)\right)L_{bg}\left(\lambda \right)\right]+L_{s}^{↑}\left(\lambda \right)+L_{atm}^{↑}\left(\lambda \right) \end{array}$$


$L_{target}\left(\lambda \right)$ is the observed at-sensor target radiance; λ is wavelenth; ε(λ) is spectral CO_2_ gas emissivity; $B_{CO_{2}}\left(\lambda ,T_{CO_{2}}\right)$ is the spectral radiance of the hot CO_2_ gas, assuming a blackbody in the Planck function with the combusted CO_2_ gas temperature ($T_{CO_{2}}$); τ(λ) is the spectral atmospheric transmittance, and $L_{s}^{↑}\left(\lambda \right)$ and $L_{atm}^{↑}\left(\lambda \right)$ are the spectral upwelling solar and thermal path radiance, respectively, reaching the sensor.

According to the spectral emissivity of combusted CO_2_ gas [38], ε(λ) is relatively high. Therefore, the term $L_{bg}$ can be removed. According to MWIR radiometric characteristics [42], the contribution of solar radiance, $L_{s}^{↑}(\lambda_{CO_{2}})$, from air scattering is very small, even for very dry conditions (less than 2% at 5 μm) [42]. Ignoring the solar term, we can simplify Equation (Equation 1) to Equation(Equation 2):
(2)$$\begin{array}{c} L_{target}\left(\lambda \right)=\tau \left(\lambda \right)\left[\varepsilon \left(\lambda \right)B_{CO_{2}}\left(\lambda ,T_{CO_{2}}\right)\right]+\left(1-\tau \left(\lambda \right)\right)B_{atm}\left(\lambda ,T_{atm}\right) \end{array}$$
where the definition of thermal upwelling, $L_{atm}^{↑}\left(\lambda \right)$, is replaced by the multiplication of blackbody radiation of the atmosphere, $B_{atm}\left(\lambda ,T_{atm}\right)$ and 1−τ(λ). The observed target spectral radiance, $L_{target}$, directly depends on both emissivity ε(λ) of hot CO_2_, and atmospheric transmittance, τ(λ), that should be analyzed specifically.

In addition, neighboring pixels to target hot CO_2_ has only atmospheric spectral path radiance at wavelenth λ. Therefore, the received background signal can be expressed with Equation (Equation 3):
(3)$$\begin{array}{c} L_{neighbor}\left(\lambda \right)=\left(1-\tau \left(\lambda \right)\right)B_{atm}\left(\lambda ,T_{atm}\right) \end{array}$$


If we apply center-surround difference by subtracting Equation (Equation 3) from Equation (Equation 2), which is the same as a mean subtraction filter (MSF) operation for a specific band image, the target-only signal can be extracted with Equation (Equation 4). This physical property is used in the D-MSF to extract the target (hot CO_2_) signature:
(4)$$\begin{array}{c} L_{targetOnly}\left(\lambda \right)=L_{target}\left(\lambda \right)-L_{neighbor}\left(\lambda \right)=\tau \left(\lambda \right)\left[\varepsilon \left(\lambda \right)B_{CO_{2}}\left(\lambda ,T_{CO_{2}}\right)\right] \end{array}$$


Conventional ships are powered by diesel engines, and the corresponding hydrocarbon combustion produces water vapor, H_2_O) and CO_2_ as expressed in Equation (Equation 5) [39,43]:
(5)$$\begin{array}{c} C_{n}H_{2n+2}+\left(3n+1\right)O_{2}\to \left(n+1\right)H_{2}O+nCO_{2} \end{array}$$


The first important parameter is spectral emissivity ε(λ) in Equation (Equation 2). The normal temperature of combusted CO_2_ gas from ships is approximately 600 K by infrared signature suppression (IRSS) [44]. The emission band-width of CO_2_ gas is proportional to the temperature of the CO_2_ gas [39]. Figure 2 demonstrates this phenomenon by changing CO_2_ temperature using the high-resolution transmission molecular absorption database (HITRAN) simulation on the web (http://hitran.iao.ru/molecule/). As the CO_2_ temperature increases, the absorption band range is broader. The absorption band range of normal atmospheric temperature (300 K) is 2300–2380 cm^−1^ (or 4.20–4.35 μm), where the inverse acts as atmospheric transmittance. The spectral unit can be either the wavelength (λ) [μm] or wavenumber (*k*) [cm^−1^] by unit selection (k=10,000/λ). For hot CO_2_, the spectral absorption coefficient is the same as the spectral emission coefficient , ε(λ), in Equation (Equation 2).

The second important parameter is spectral atmospheric transmittance τ(λ) in Equation (Equation 2). In the MODTRAN simulation in the MWIR band, the spectral transmittance of the CO_2_ band (2320–2375 cm^−1^ or 4.21–4.31 μm) decreases abruptly with distance, as shown in Figure 3. The average transmittance in the CO_2_ band is 0.13, 0.005, and 0 at 5, 20, and 100 m, respectively. Note that the atmospheric transmittance band of zero value coincides with the spectral absorption coefficient of CO_2_ at 300 K in the top part of Figure 2. In addition, it remains only the atmospheric path radiance in Equation (Equation 2) in the CO_2_ absorption band, τ(λ)=0), where there is no target or background signals.

Multiplication of the spectral emissivity of hot CO_2_ and atmospheric transmittance produces double spikes, as shown in the bottom portion of Figure 1. The spectral feature where a hot CO_2_ emission band is broader than that which is absorbed by normal atmospheric CO_2_ generates CO_2_ double spikes. The midwave spectral information is acquired via TELOPS MWIR hyperspectral camera [40]. It can provide calibrated spectral radiance images with a high spatial and spectral resolution from a Michelson interferometer in the short-wave to midwave bands (1.5–5.6 μm).

### 2.2. CO_2_-DS-Based Ship Plume Detection

Motivated by the double-spike phenomenon of hot CO_2_, a novel ship-detection method is proposed, as shown in Figure 4. The proposed method is called CO_2_-DS because it is based on the carbon dioxide-double spike feature. The proposed CO_2_-DS approach consists of three steps: spectral band selection, spatial small-target detection, and the final ship detection. The spectral band selection sets the search range in the spectral domain focusing on the CO_2_ absorption band. Then, each CO_2_ spike is detected by a carefully designed spatial filter (the directional-mean subtraction filter). Finally, the target ship is confirmed using a joint detection operation.

In Step 1, the specific spectral band range should be defined to attain successful remote-ship detection with extremely few false positives. The proposed CO_2_-DS method is based on the atmospheric CO_2_ absorption band (2320–2375 cm^−1^ or 4.21–4.31 μm) that should be included. In addition, double-spike bands emitted by hot CO_2_ should be included. The first spike range is below 2300 cm^−1^, and the second spike range is above 2380 cm^−1^.

The details of Step 2 and Step 3 in Figure 4 are described in Figure 5. Each spike signal is detected by a parallel search algorithm: the first spike detection and the second spike detection.

The starting wavenumbers are 2300 cm^−1^ and 2380 cm^−1^ because there is no signal at all except for atmospheric path radiance. Given a wavenumber, a test image of this band is probed by D-MSF-based small infrared target detection [4]. D-MSF is adopted in this paper because it is a well-proven method and robust against thermal noise in the maritime environment. Figure 6 summarizes the detailed flow of the D-MSF graphically. Given a hypothesized spectral band image, $I_{\lambda }\left(x,y\right)$, a mean subtraction filter, h(x,y), is used to enhance the signal-to-clutter ratio (1). Then, directional local median $I_{\lambda }^{D}\left(x,y\right)$ is used to estimate row directional background (2) from the MSF result, $I_{\lambda }^{MSF}\left(x,y\right)$. Subtracting (2) from (1) in Figure 6 produces a clearer signature image. The region of interest (ROI) is generated by an initial thresholding (th1) and eight-nearest neighbor-based clustering. Final detection is achieved by applying adaptive thresholding th2 in constant false alarm rate (CFAR) detector to the subtracted image with the given ROI. It can provide a signal-to-clutter ratio (SCR) as well as the ROI of the target.

If there is no detection, the wavenumber ($k_{1}$) of the first spike is decreased by Δ, and that of the second spike is increased by Δ, the value of which is 13 cm^−1^ in this paper. If there is a detection, each neighboring band image is probed additionally to obtain a maximum signal score.

Finally, in Step 3, the joint detection process is performed from the first and second spike detections: [$x_{1},y_{1},w_{1},h_{1},SCR_{1}$], [$x_{2},y_{2},w_{2},h_{2},SCR_{2}$], where [·] denotes [column, row, width, height, SCR] of the detected target. If two detections exist, then the final target information is merged with Equation (Equation 6). If the merged SCR is larger than a predefined threshold (th3), then it is declared a final target.
(6)$$\begin{array}{c} Final\;target\;info=\left[\frac{x_{1}+x_{2}}{2},\frac{y_{1}+y_{2}}{2},\frac{w_{1}+w_{2}}{2},\frac{h_{1}+h_{2}}{2},\left(SCR_{1}+SCR_{2}\right)\right] \end{array}$$


Figure 7 shows the spectral profile of a remote ship’s plume and an enlarged view focusing on the atmospheric CO_2_ absorption band. The numbers (1), (2), (3), and (4) represent the probing band locations of the proposed CO_2_-DS method; (1) and (2) belong to the first spike band, and (3) and (4) belong to the second spike band. Note that the first spike is not clear in remote cases but it does not matter in our CO_2_-DS method since D-MSF is used. Figure 8 shows the corresponding band images, and the max SCR is used to select a band image for each spike: (2) and (3) are selected. The bottom image in Figure 8 represents the result of joint detection from the detection results of the first spike and the second spike.

## 3. Experimental Results

### 3.1. Experiment 1: Analysis of Signature Variation

MWIR hyperspectral images of remote ships were acquired with the open path TELOPS Hyper-Cam Mid-Wave Extended (MWE) model [40]. It could provide calibrated spectral radiance images with a high spatial and spectral resolution from a Michelson interferometer in the shortwave to midwave band (1.5–5.6 μm). The spatial image resolution was 320 × 256, and the spectral resolution was up to 0.25 cm^−1^. The noise-equivalent spectral radiance (NESR) was 7 $\left[\mathrm{nW}/\left({\mathrm{cm}}^{2}\cdot \mathrm{sr}\cdot {\mathrm{cm}}^{-1}\right)\right]$, and the radiometric accuracy was approximately 2 K. The field of view was 6.5 × 5.1 degrees.

In the first experiment, the effect of signature variation was compared, depending on target distance. Figure 9 shows the spectral profiles of hot CO_2_ plumes, both near (78 m) and remote (2 km). A strong double-spike phenomenon can be observed for near CO_2_, as shown in Figure 9 (left side) due to relatively weak signal attenuation. However, a remote target does not show such a distinctive double-spike pattern as seen in Figure 9 (right side), which makes it clear that only the spectral profile-based approach was not effective at detecting targets. This conclusion was confirmed through a real experiment applying the well-known spectral angle mapper-based detector [45] to the spectral profiles. Figure 10 shows the spectral profile-based detection results. The left column is the training region; the middle column is a representative spectral profile after training (just the mean of spectral profiles); and the right column is SAM-based detection results indicated by red dots. For the near target, the hot CO_2_ plume region could be detected correctly with a similarity threshold of 0.8, as shown in Figure 10a. On the other hand, the remote target signature was unclear, which led to false positive detections with a similarity threshold of 0.98, as shown in Figure 10b.

### 3.2. Experiment 2: Detection Parameter Analysis

In the second experiment, key parameters of the proposed CO_2_-DS method were analyzed and evaluated. There are four parameters; spectral band range for probing, initial threshold (th1) and CFAR threshold (th2) in the D-MSF, and final threshold (th3) from joint detection. The spectral band range for robust ship detection should be determined analytically based on Figure 11. Basically, the atmospheric CO_2_ absorption band should be included. Since the air temperature is usually 300 K, this band is fixed at [2320–2380 cm^−1^] as shown in the bottom part of Figure 11. The first-spike band starts at 2300 cm^−1^ and ends at 2180 cm^−1^ considering band offset. The second-spike band starts at 2380 cm^−1^ and ends at 2450 cm^−1^ considering band offset. The band images below 2180 cm^−1^ or above 2450 cm^−1^ show background clutter with target information that should be excluded to ensure extremely few false positives.

The selection of threshold1 (th1) is used to determine an ROI in order to investigate whether it is a target or not, as shown in Figure 6. Figure 12 shows the image for $I_{\lambda }^{MSF}\left(x,y\right)$, where the bright blob represents the hot CO_2_ target, and others are background pixels. The 3D surf views of the target patch and background patch show the signal levels, and the lower left graph of Figure 6 displays the signal intensity profile corresponding to the dotted red line in the image. Note that the max signal is $20\times 10^{-5}\left[{W/m}^{2}{,\mathrm{sr},\mathrm{cm}}^{-1}\right]$ and the max background signal is $5\times 10^{-5}\left[{W/m}^{2},{\mathrm{sr},\mathrm{cm}}^{-1}\right]$. Considering the safety margin of factor 2, th1 was set at $1.0\times 10^{-4}\left[{W/m}^{2},{\mathrm{sr},\mathrm{cm}}^{-1}\right]$. The threshold 2 parameter (th2) can be set by analyzing the SCR values for different targets, as shown in Figure 13. Most SCRs of a spike band image are larger than 12.8, so, th2 can be set at 10.0 for a safe detection capability. The last threshold (th3) was varied to calculate the receiver operating characteristic (ROC) curve in order to compare different target-detection methods.

### 3.3. Experimental 3: Performance Evaluation

Based on these parameter analyses, the final detection performance was evaluated by comparing the proposed CO_2_-DS, D-MSF [4], and the high-boost multi-scale contrast measure (MLCM) [46]. MLCM is based on the human visual system, and greatly improves the detection rate for small infrared targets. The test images consist of 53 hypercubes acquired by a TELOPS MWIR hyperspectral camera. Three different ships with distance ranges between 1.5 km and 4 km were considered. For the baseline methods, test images were prepared by making broad band images in the spectral range of 3.5–5.6 μm. The ROC metric was used for a fair detection performance based on false positives per image (FPPI). Figure 14 summarizes detection performance. The proposed method (the red solid line) showed an ideal detection curve. MLCM ranked second, and D-MSF showed the worst result. Visual inspection was conducted at the indicated FPPI (0.65/image). With a normal homogeneous background, the three methods showed reasonable working performance, as seen in Figure 15. Magenta circles represent ground truth, cyan rectangles are detection results from the proposed CO_2_-DS, and yellow rectangles represent baseline detection results. If there was strong cloud clutter, D-MSF generated many false alarms, as shown in Figure 16. In a strong sea-clutter environment, MLCM detected a lot of sea-glints, as shown in Figure 17. Note that the proposed CO_2_-DS showed stable detection results without any false positives from sky or sea clutter. In addition, the ship was located at 4 km (near the horizon) and was detected successfully by our proposed method.

### 3.4. Experiment 4: Detection Performance Analysis

It can be useful to analyze the effect of weather condition and target distance to the detection performance. The proposed CO_2_-DS method is based on Equation (Equation 4), especially atmospheric transmittance (τ(λ)). Therefore, if we use the MODTRAN-based atmospheric transmittance calculation and measured background noise, we can conduct the Monte Carlo simulation. Figure 18 represents the MODTRAN-based atmospheric transmittance calculation by changing distance at a specific wavenumber (2390.16 cm^−1^) or wavelength (4.184 μm). The atmosphere model was selected either “MidLatitude Summer” or “MidLatitude Winter” to consider the effect of humidity and temperature. The distance length changed from 0.1 km to 6.0 km with 0.1 km interval.

The proposed CO_2_-MS detector uses target-background difference method (mean subtraction filter). Therefore, the path radiance term can be removed. Figure 19 shows a filtered map for the spectral band image at 2393 cm^−1^. The maximum target signal-background at 2 km is $6.1742\times 10^{-4}\left[{W/m}^{2}\cdot \mathrm{sr}\cdot {\mathrm{cm}}^{-1}\right]$. If one uses transmittance of 0.18 at 2 km, the original signal-background is $3.4\times 10^{-3}\left[{W/m}^{2}\cdot \mathrm{sr}\cdot {\mathrm{cm}}^{-1}\right]$. Background noise can be modeled as the rectified Gaussian noise model with 0 mean and standard deviation of $1.3964\times 10^{-5}\left[{W/m}^{2}\cdot \mathrm{sr}\cdot {\mathrm{cm}}^{-1}\right]$. Th2 (SCR) is set as 10 in this simulation. Figure 20 shows the atmospheric transmittance and Monte Carlo simulation-based detection performance according to the target distance and weather conditions. Normally, the transmittance in mid-latitude summer is lower than that in mid-latitude winter because of higher water vapor contents (higher humidity), which leads to shorten the detection range as Figure 20b. If we fix target distance at 5 km, the detection rate is reduced from 95.4% to 5.2%.

The proposed CO_2_-MS detection method shows extremely low false positives in ship detection. In addition, CO_2_-MS is based on the thresholding and clustering. So, it can detect multiple targets. Because the CO_2_-MS can use both the spectral and spatial information, the size of the ship (fishing boat, military ship, or cargo ship) can be identified if each ship has different type of fuel, engine, gas volume, and temperature. However, it has several limitations such as expensive spectral measurement device, weak to weather condition such as high humidity.

## 4. Conclusions

Remote-ship detection is important in various applications, such as navigation and surveillance. Midwave infrared image-based ship detection is deployed due to its long-range detection capability in the maritime environment. However, MWIR images show strong response to cloud clutter and sea glint, which generate many false positives, even in state-of-the-art methods. This paper proposed a completely different approach by fusing spectral and spatial information in the MWIR carbon dioxide absorption band with the double-spike phenomenon. The atmospheric CO_2_ absorption band (2320–2375 cm^−1^) can block all background radiance, and remote hot CO_2_ can penetrate through the spectral double-spike band, which is detected by the proposed carbon dioxide-double spike algorithm. Experimental results in a real cluttered sea environment with remote ships showed excellent detection performance with extremely few false positives if the ships’ diesel engines are working.

## Figures and Tables

**Figure 1 sensors-20-02896-f001:**
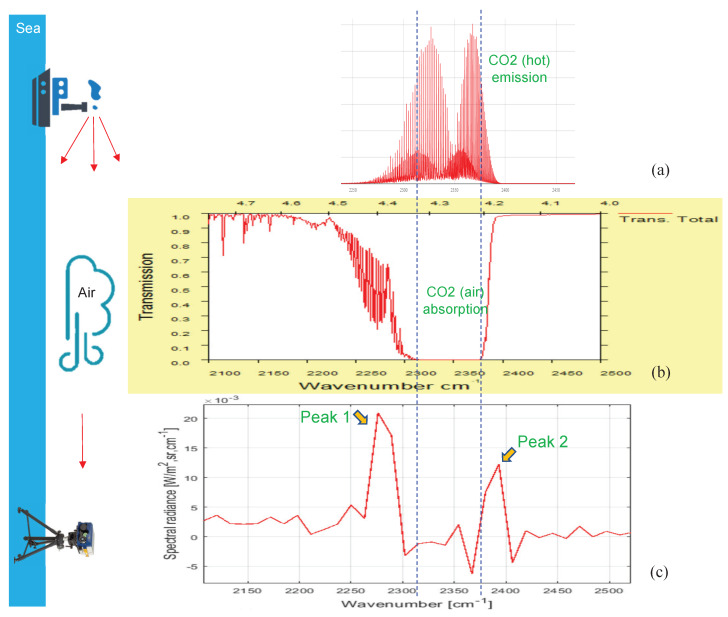
Operational concept of remote ship detection and the double spike-signature generation mechanism of hot carbon dioxide: (**a**) spectrum of a hot CO_2_ emission by a ship, (**b**) spectral atmospheric transmittance, and (**c**) received spectral radiance of hot CO_2_ from a ship’s plume.

**Figure 2 sensors-20-02896-f002:**
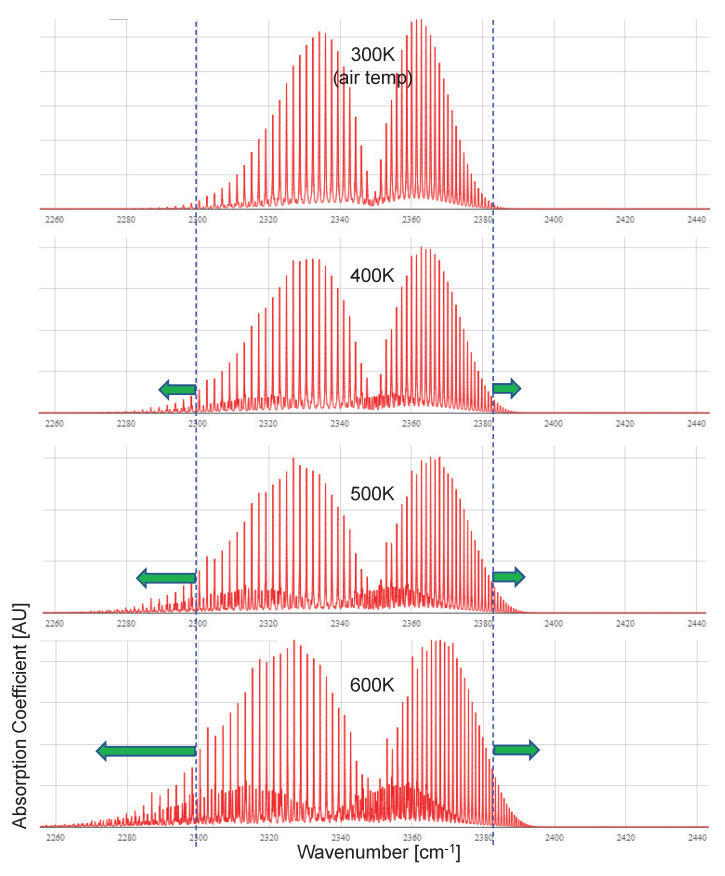
Spectral absorption coefficient variations of carbon dioxide according to different gas temperatures.

**Figure 3 sensors-20-02896-f003:**
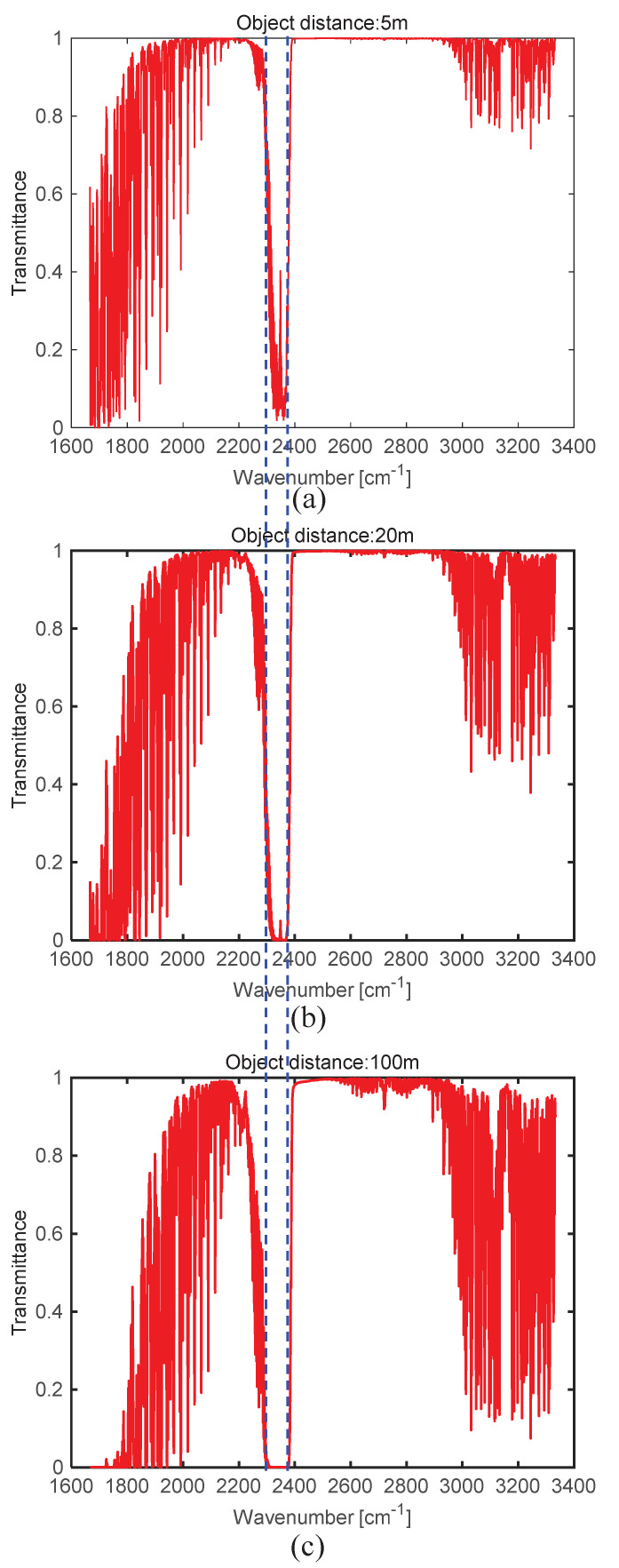
Spectral transmittance according to ship-camera distance in the MWIR band: (**a**) 5, (**b**) 20, and (**c**) 100 m.

**Figure 4 sensors-20-02896-f004:**
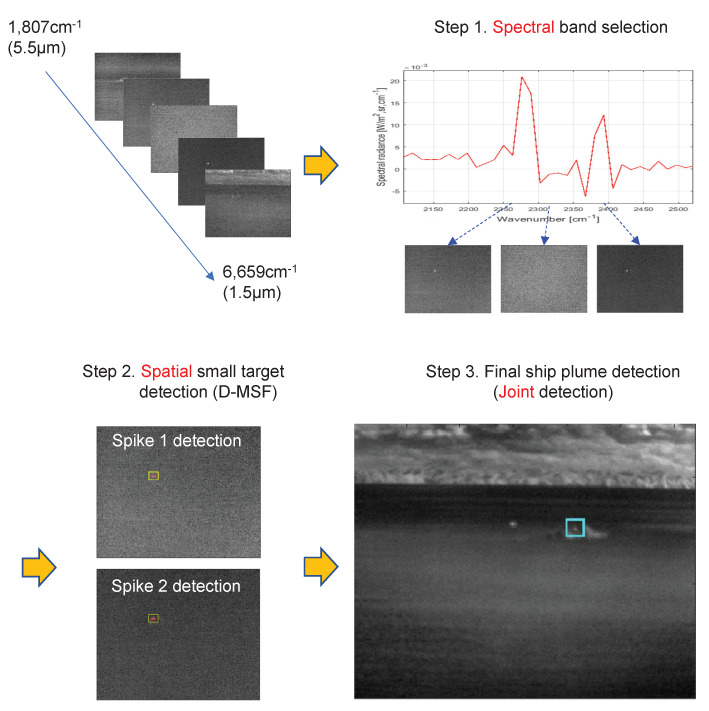
Overall processing flow of remote-ship detection using spectral–spatial information of hot CO_2_ plumes.

**Figure 5 sensors-20-02896-f005:**
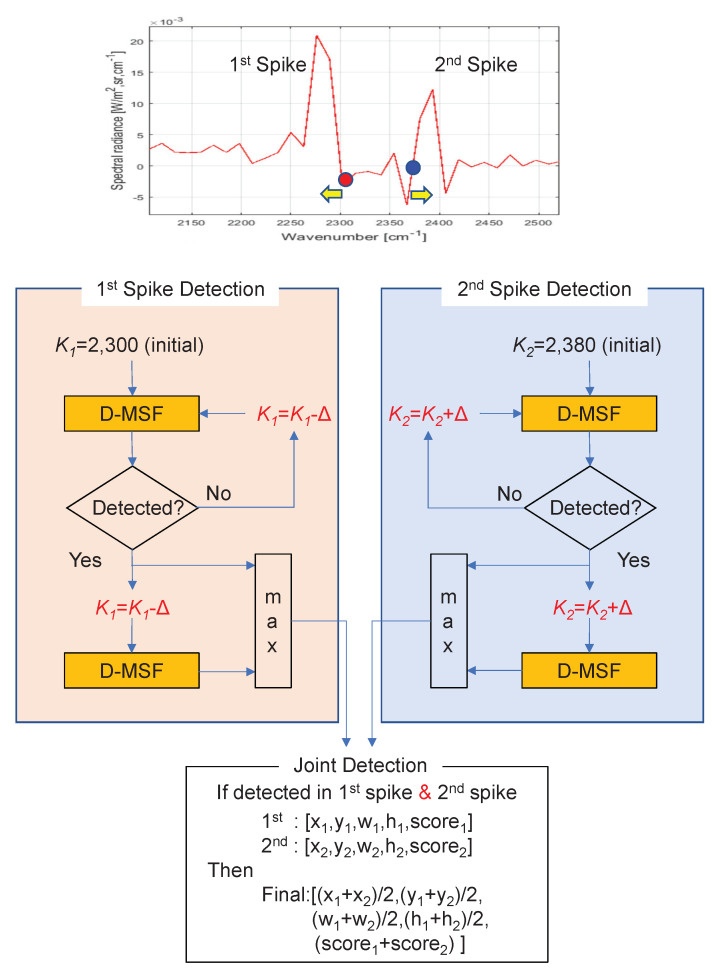
Details of the proposed CO_2_-DS: first spike detection, second spike detection, and joint detection.

**Figure 6 sensors-20-02896-f006:**
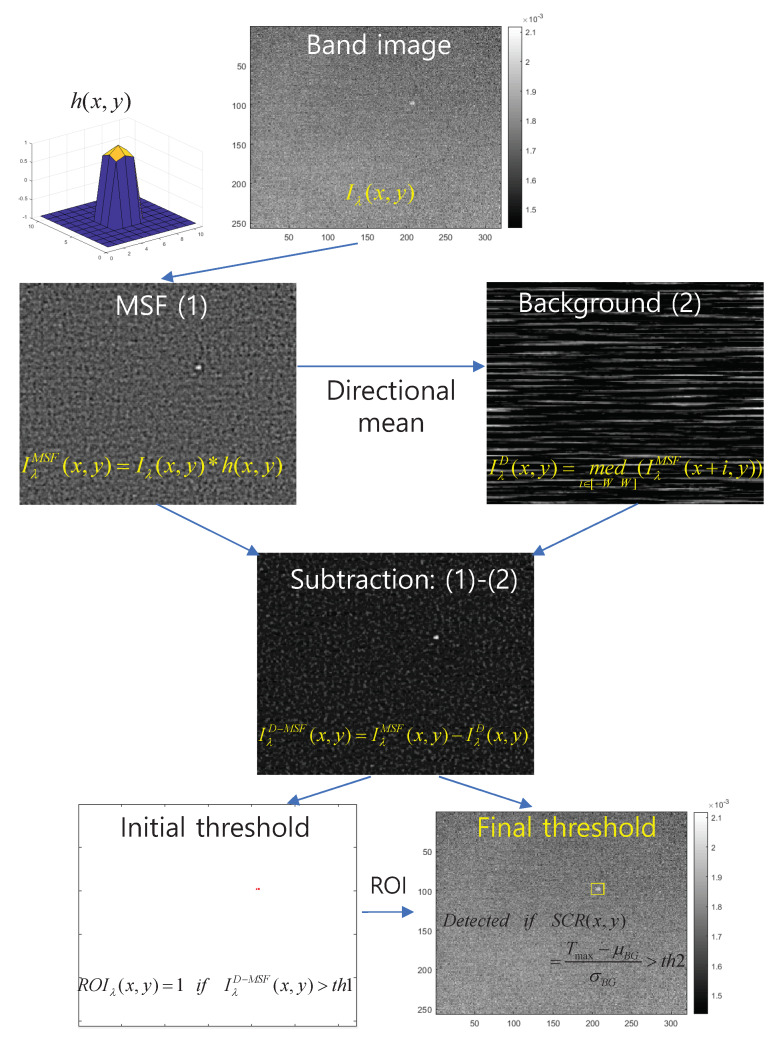
Operational flow of the directional-mean subtraction filter (D-MSF) for small-target detection in a band image.

**Figure 7 sensors-20-02896-f007:**
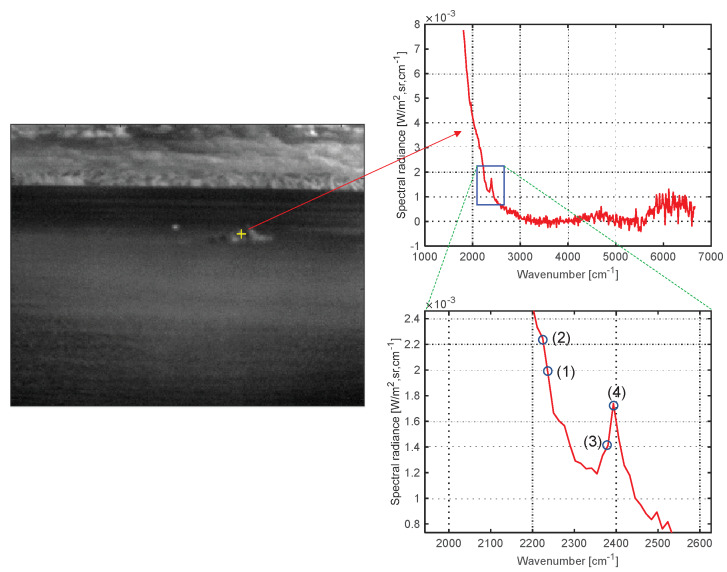
Spectral profile of a remote ship’s plume with the probed band positions: first spike: (1) and (2); second spike: (3) and(4).

**Figure 8 sensors-20-02896-f008:**
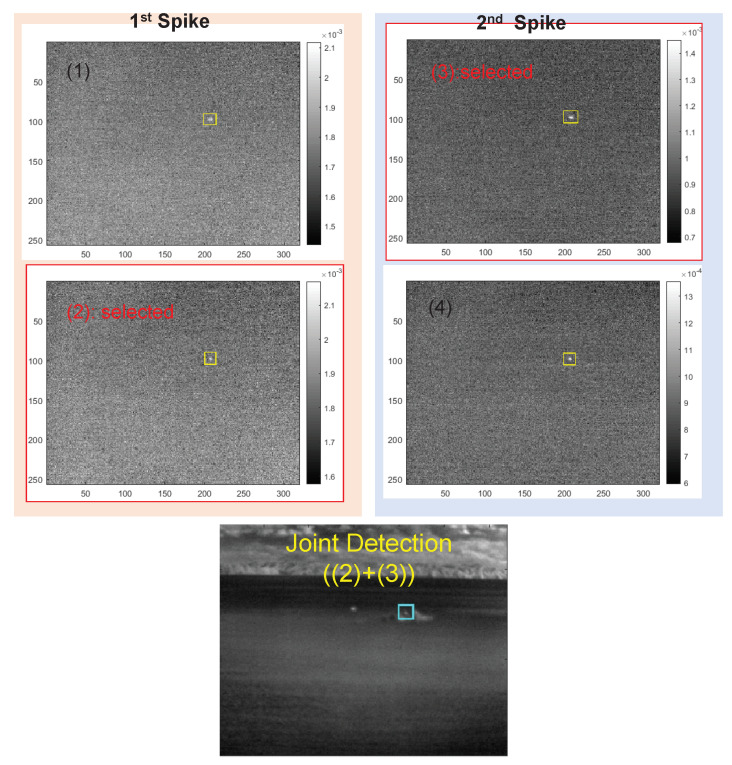
Probed band images of first and second spikes corresponding to (1) and (2), and (3) and (4), respectively. The final target is detected via joint detection from the selected max signature for each spike.

**Figure 9 sensors-20-02896-f009:**
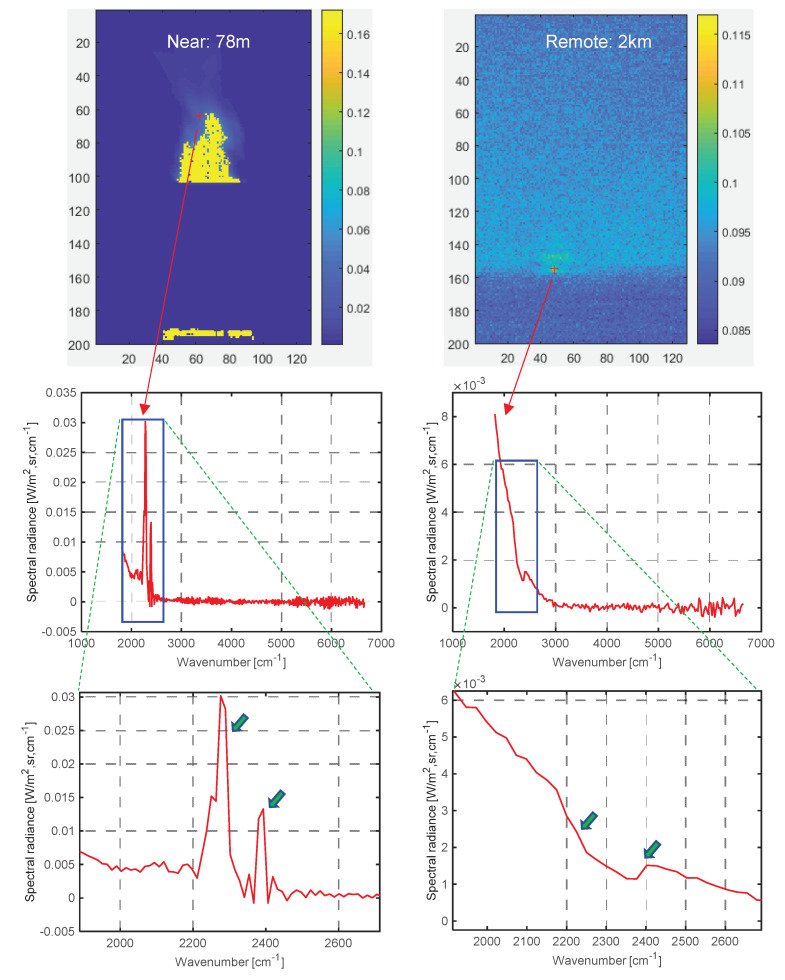
Comparison of CO_2_ signatures in terms of distance: (left) near CO_2_ (78 m), (right) remote CO_2_ (2 km).

**Figure 10 sensors-20-02896-f010:**
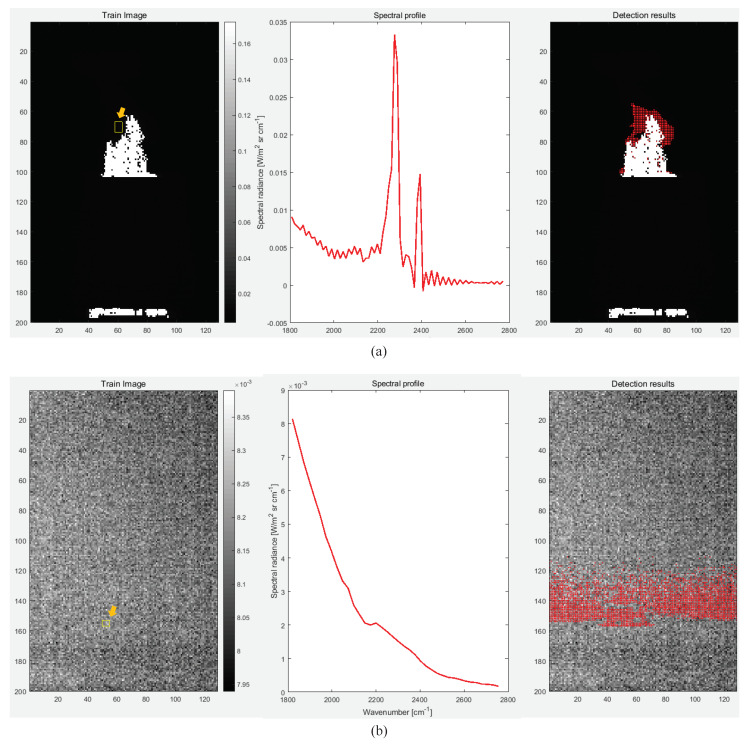
Spectral profile-based target detection results: (**a**) near CO_2_, (**b**) remote CO_2_.

**Figure 11 sensors-20-02896-f011:**
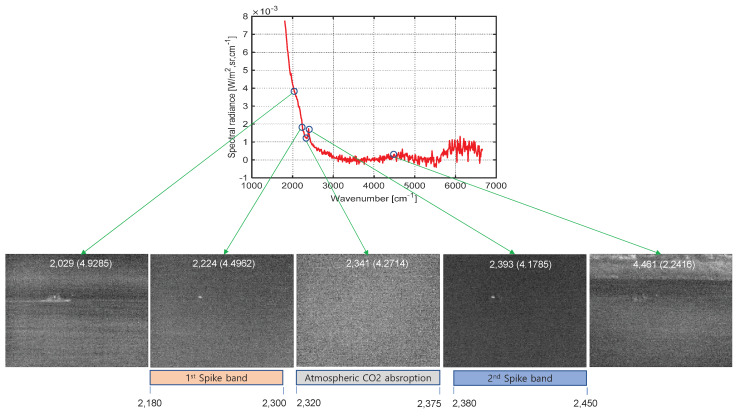
Analysis of spectral band ranges for ship detection by visualizing corresponding band images.

**Figure 12 sensors-20-02896-f012:**
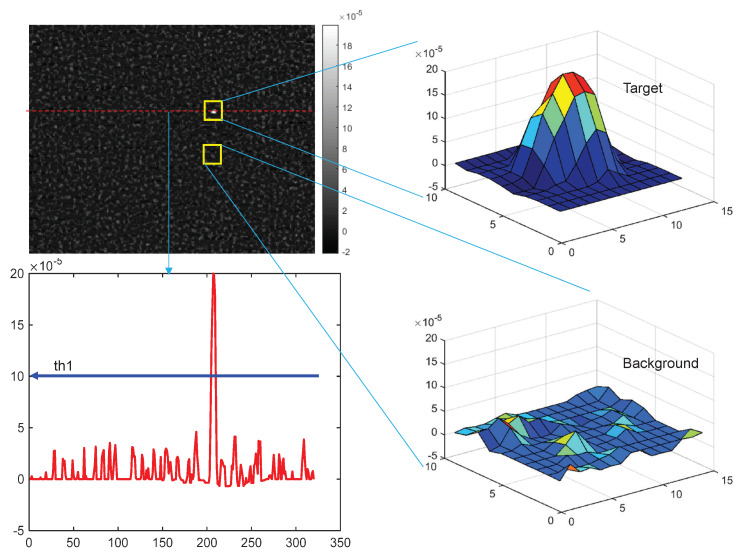
Parameter analysis of threshold 1 (th1) from directional mean subtraction image.

**Figure 13 sensors-20-02896-f013:**
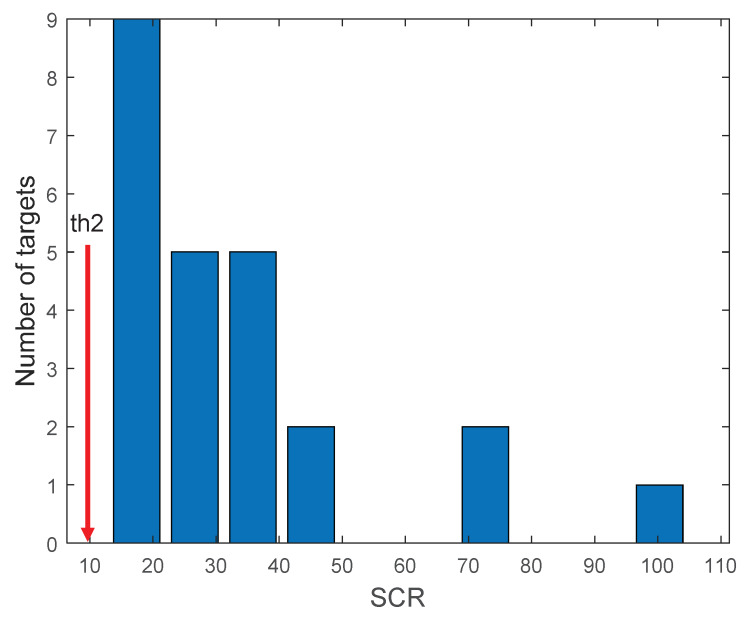
Parameter analysis of threshold 2 (th2) from statistics of SCR values of different targets.

**Figure 14 sensors-20-02896-f014:**
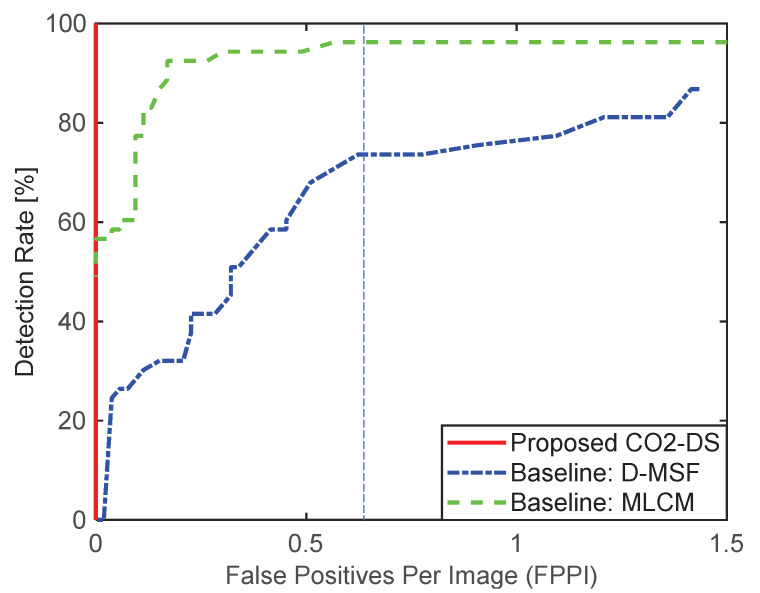
Receiver operating characteristic (ROC)-based comparison of detection performance.

**Figure 15 sensors-20-02896-f015:**
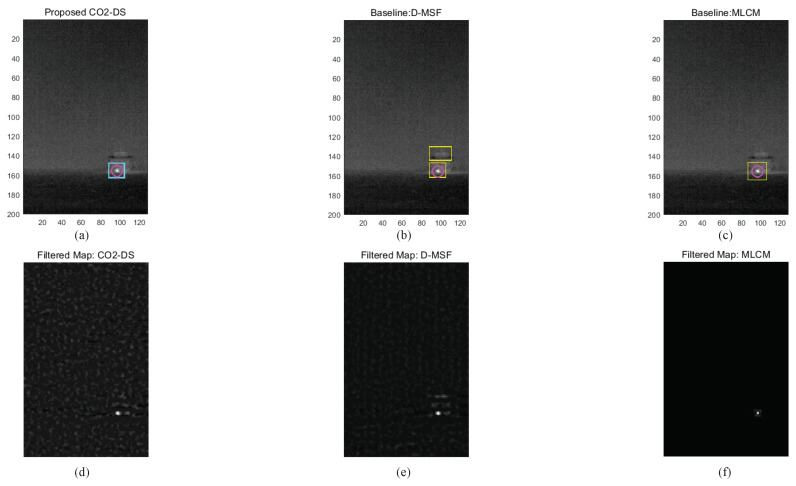
Comparative detection examples for a homogeneous background: (**a**–**c**) detection results, (**d**–**f**) filtered maps.

**Figure 16 sensors-20-02896-f016:**
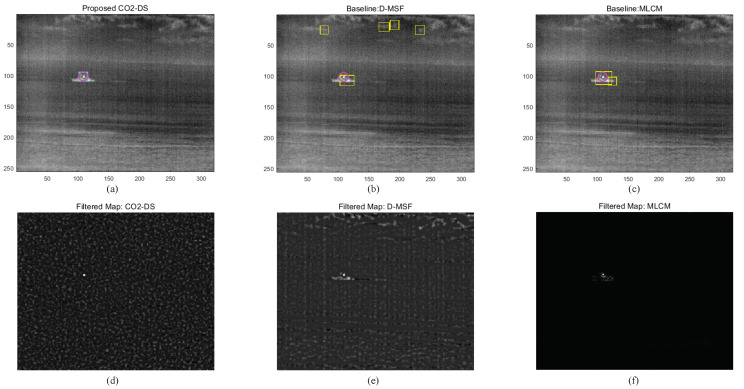
Comparative detection examples for cloud-clutter backgrounds: (**a**–**c**) detection results, (**d**–**f**) filtered maps.

**Figure 17 sensors-20-02896-f017:**
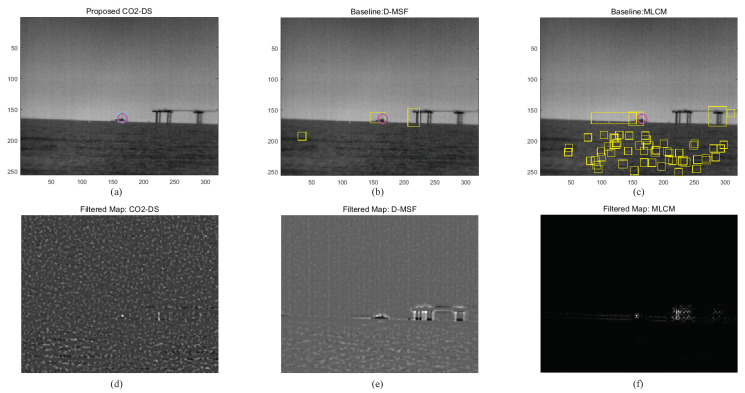
Comparative detection examples for sea-clutter backgrounds: (**a**–**c**) detection results, (**d**–**f**) filtered maps.

**Figure 18 sensors-20-02896-f018:**
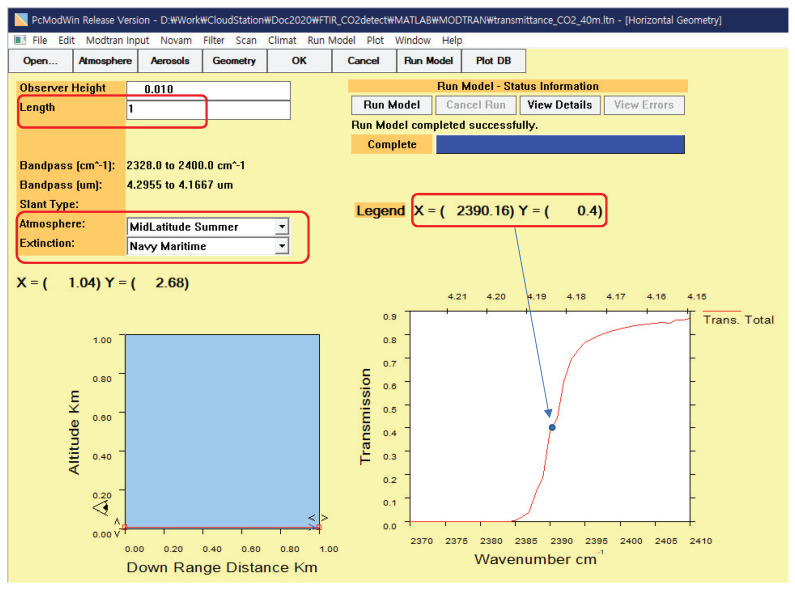
Moderate-Resolution Atmospheric Radiance and Transmittance (MODTRAN) environment for atmospheric transmittance calculation.

**Figure 19 sensors-20-02896-f019:**
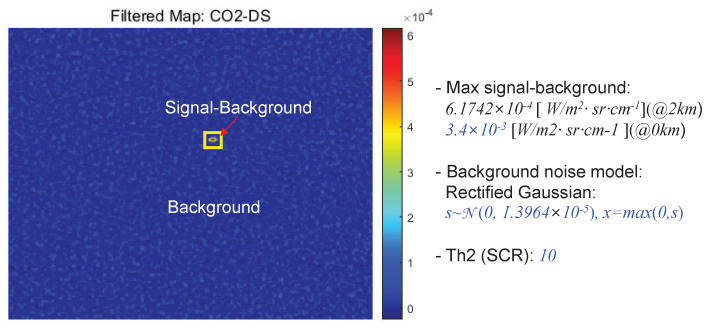
MODTRAN environment for atmospheric transmittance calculation.

**Figure 20 sensors-20-02896-f020:**
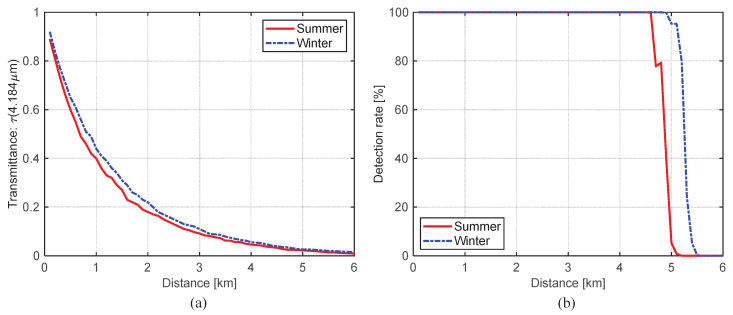
(**a**) Atmospheric transmittance vs. distance, (**b**) Monte Carlo simulation-based detection results.

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
