# Peer review of "Extremely Robust Remote-Target Detection Based on Carbon Dioxide-Double Spikes in Midwave Spectral Imaging"

_sensors, 2020, doi:10.3390/s20102896_

Round 1

Reviewer 1 Report

Comments to the Author

Remote-target detection based on carbon dioxide double spikes in midwave spectral imaging is definitely an interesting and highly promising method for remote-target detection of ships. The approach described by the authors is quite reasonable.

However, I have several questions for this work.

1) According to my understanding, the infrared spectrum is greatly affected by humidity. The authors are suggested to include comments on the different humid conditions and their implications on the recognition rate of ship-target detection.

2) For remote-target detection, distance is the main element for discussion. In this study, the authors have commented on the situations when the target is at 78m and 2km. I want to know the rate of ship-target detection when different distances are used with the present method.

Consequently, the authors are suggested to list the distances where this method works and fails. Any disadvantages of the present approach also should be illustrated.

3) I noticed that the authors have detected only one target, but in reality, the number of targets is usually more than one. The authors are suggested to make some comments on the multi-target situation.

4) The amount of CO2 emitted by ships is different. The authors are suggested to comment that the size of the ship whether can be identified based on the carbon dioxide double spikes in midwave spectral imaging.

5) Last but not least, at present, the content of this paper would be quite difficult to understand for ordinary readers, especially for the researchers without midwave spectra or infrared background. For practical applications and in order to make the approach more understandable, it is highly desirable to provide some example script which would allow to test this method. One possible way to do this would be to write an example, in Matlab code and submit it as supporting information.

I believe that carbon dioxide double spikes is a quickly developing approach, and many works on this topic will appear in the near future. As of this submission, I still consider it interesting despite the above comments, and in case if the authors can provide a scientifically sound answer, I would recommend to accept it.

Reviewer 2 Report

This paper proposes an approach, called carbon dioxide-double spike (CO2-DS) detection in midwave spectral imaging for infrared ship-target detection in the maritime environment. In general, the article is well written, well organized, with in-depth theoretical analysis of physical mechanism and rich experimental verification. In my opinion, the article can be accepted after minor revision. 

My minor concern is that in the introduction part, more background of marine ship detection should be mentioned, such as SAR ship detection. Correspondingly, some references on SAR ship detection should be cited, such as:

(1) “Squeeze and excitation rank faster R-CNN for ship detection in SAR images,” IEEE Geosci. Remote Sens. Lett., vol. 16, no. 5, pp. 751-755, May 2019.

(2)“DRBox-v2: An Improved Detector With Rotatable Boxes for Target Detection in SAR Images,” IEEE Transactions on Geoscience and Remote Sensing, vol. 57, no. 11, pp. 8333-8349, 2019.

(3) “Contextual region-based convolutional neural network with multilayer fusion for SAR ship detection,” Remote Sensing , vol. 9, no. 8, pp. 860, 2017 and etc.

Reviewer 3 Report

Dear authors,

Thank you for submitting this interesting paper.

I consider it as amost ready for publications.  However I pointed some few elements to be updated or improved.

The abstract shall be improved on the following points:

  • you refer to conventional method: you should cite some of them here
  • you should clarify the context of the paper. This applies for observation from the ground/shore/vessel and or airborne observation. Please clarify

Line 60 "both types of detection", you should reword it to clarify that this refer to the detection of both spikes. For the detection of each spike you apply the same detection technique (or similar) thus the same detection type.

Line 79, please provide reference to TELOPS.

Line 161 and following: the (1),(2) refer to figure 6. Please make it more explicit.

Lines 182 to 163: why don't you perform detection in the spectral domain ? As you explain that there is a double spike, then this correspond to a valey between the two spikes. A detector of a valey between two spikes should be much simpler.

Figure 7: type: second spike (not spiek)

Line 216: is it consistes or consist ?

Round 2

Reviewer 1 Report

I appreciate a big work made by the Authors since the first version. This revision of a manuscript is very well written.
With the script provided in Supplementary, the suggested approach can be verified and tested.
I am sure this work will be highly appreciated and highly cited.